# Evaluation of the Potential Role of Proprotein Convertase Subtilisin/Kexin Type 9 (PCSK9) in Niemann–Pick Disease, Type C1

**DOI:** 10.3390/ijms21072430

**Published:** 2020-03-31

**Authors:** Niamh X. Cawley, Anna T. Lyons, Daniel Abebe, Christopher A. Wassif, Forbes D. Porter

**Affiliations:** 1Section on Molecular Dysmorphology, Division of Translational Medicine, Eunice Kennedy Shriver National Institute of Child Health and Human Development, National Institutes of Health, Department of Health and Human Services, Bethesda, MD 20892, USA; anna.lyons@nih.gov (A.T.L.); wassifc@cc1.nichd.nih.gov (C.A.W.); 2Research Animal Management Branch, National Institute of Child Health and Human Development, National Institutes of Health, Department of Health and Human Services, Bethesda, MD 20892, USA; abebed@mail.nih.gov

**Keywords:** Niemann–Pick C, PCSK9, neurodegeneration, VLDLR, ApoER2, lysosomal storage, NPC1 KO mouse

## Abstract

Niemann–Pick disease, type C1, is a cholesterol storage disease where unesterified cholesterol accumulates intracellularly. In the cerebellum this causes neurodegeneration of the Purkinje neurons that die in an anterior-to-posterior and time-dependent manner. This results in cerebellar ataxia as one of the major outcomes of the disease. Proprotein convertase subtilisin/kexin type 9 (PCSK9) plays a significant role in the regulation of serum cholesterol levels by modulating LDL receptor levels on peripheral tissues. In the central nervous system, PCSK9 may have a similar effect on the closely related VLDL and ApoE2 receptors to regulate brain cholesterol. In addition, regulation of VLDLR and ApoER2 by PCSK9 may contribute to neuronal apoptotic pathways through Reelin, the primary ligand of VLDLR and ApoER2. Defects in reelin signaling results in cerebellar dysfunction leading to ataxia as seen in the *Reeler* mouse. Our recent findings that *Pcsk9* is expressed ~8-fold higher in the anterior lobules of the cerebellum compared to the posterior lobule X, which is resistant to neurodegeneration, prompted us to ask whether PCSK9 could play a role in NPC1 disease progression. We addressed this question genetically, by characterizing NPC1 disease in the presence or absence of PCSK9. Analysis of double mutant *Pcsk9^-/-^/Npc1^-/-^* mice by disease severity scoring, motor assessments, lifespan, and cerebellar Purkinje cell staining, showed no obvious difference in NPC1 disease progression with that of *Npc1^-/-^* mice. This suggests that PCSK9 does not play an apparent role in NPC1 disease progression.

## 1. Introduction

Niemann–Pick disease, type C (NPC) is an autosomal recessive, lysosomal storage disorder that affects ~1:120,000 births [1,2]. Patients with NPC initially present with hepatosplenomegaly that is followed by progressive neurodegeneration leading to cerebellar ataxia and dementia [1,3] with patients typically succumbing to the disease in the second decade of life [4]. NPC is caused by mutations in NPC1 or NPC2, which encode for proteins that are involved in the transport of unesterified cholesterol out of the lysosome [5,6,7,8]. Defects in either of these proteins result in the accumulation of unesterified cholesterol and other sphingolipids in the lysosome [9,10]. A hallmark of the disease is the presence of neuroinflammation in the central nervous system [11,12,13,14], leading to the anterior-to-posterior patterned cerebellar Purkinje neuron death within the cerebellum [15]. The loss of Purkinje neurons is consistent with cerebellar ataxia, a major clinical finding in NPC. While much is known about the cause of NPC and the disease progression, there remains a need for a greater understanding of the molecular processes and pathways involved in the patterned neurodegeneration of the Purkinje neurons in order to develop new therapies. 

Recently we performed RNA-sequencing analysis to examine the gene expression profiles of the cerebellar lobules of 4.5-week old *Npc1^-/-^* mice [16]. This was done to identify genes and cellular pathways involved in the Purkinje neuron degeneration and potential protective expression patterns in lobule X of the cerebellum prior to disease progression. Our data identified several novel pathways of interest, including calcium, dopamine, and glutamate signaling, that may contribute to the lobule-specific susceptibility of Purkinje neurons to degeneration during disease progression [16]. One gene of interest was *Pcsk9*, that encodes proprotein convertase subtilisin/kexin type 9 (PCSK9) [17]. This gene is expressed almost 8-fold higher in the anterior lobules of both *Npc1^+/+^* and *Npc1^-/-^* cerebella compared to lobule X (Appendix A), suggesting a possible functional and lobule specific role for PCSK9 in the general physiology of the cerebellum and possibly of Purkinje neuron health during NPC disease progression.

PCSK9 is the ninth member of the proprotein convertase family of processing enzymes [18,19,20]. Members of this family of enzymes are typically involved in the basic residue specific processing of proteins within the secretory pathway of cells to generate mature secreted biologically active peptides. Proprotein convertases are synthesized as pro-enzymes in the endoplasmic reticulum (ER) and require proteolytic activation by removal of their pro-domain in the trans Golgi network (TGN) or immature and mature secretory granules. However, PCSK9 appears to be unique in that its proteolytic function is inhibited through the tight binding of its cleaved pro-domain to its catalytic domain, rendering the enzyme inactive as a secreted proteinase [21]. Hence, PCSK9 does not function as a protein processing enzyme.

The role of PCSK9 in the regulation of serum cholesterol has been extensively studied. PCSK9 associates with the extracellular EGF-like binding domain of the low-density lipoprotein receptor (LDLR) in the liver [22]. After endocytosis of the LDLR, the bound PCSK9 triggers degradation of the internalized complex within the lysosome, thus reducing the levels of LDLR present on the plasma membrane [23,24]. Such reduction of LDLR in the liver is sufficient to increase circulating levels of cholesterol. Hence, gain of function and loss of function mutations in *PCSK9* have been found to modulate serum cholesterol levels in humans [25,26,27,28,29] making PCSK9 inhibitors an attractive target to treat hypercholesterolemia in humans [30]. During development in the mouse, *Pcsk9* is expressed in the liver, kidney and intestine and in the brain it is expressed early in the telencephalon (embryonic day (E) 12) and later in the cerebellum (E17), with evidence it may be involved in neurogenesis [17,31]. In the mouse, expression of *Pcsk9* in the cerebellum persists during perinatal development, however, in the adult brain, overall levels are reduced with some signal in the external granule layer of the cerebellum [17]. Such a function appears to be essential in zebrafish development, since reduced *Pcsk9* expression in zebrafish results in significant abnormal neuronal development [31], however, this is in contrast to the mouse model, since *Pcsk9^-/-^* mice appear to develop normally [32,33].

The function of PCSK9 in the central nervous system (CNS) is not fully understood despite implications for a role in Alzheimer disease (for review see [34]). In the brain, two receptors closely related to LDLR; very low-density lipoprotein receptor (VLDLR) and apolipoprotein E receptor-2 (ApoER2), are expressed and have been studied as possible targets of PCSK9. PCSK9 and a gain of function mutant of PCSK9 enhanced cellular degradation of these receptors [35], and PCSK9 binds to LDLR, VLDLR, and ApoER2 with similar sub-micromolar binding constants [36]. Indeed, it has been proposed that PCSK9 modulates neuronal apoptotic signaling pathways via regulation of ApoER2 levels in the brain [37]. VLDLR and ApoER2 are implicated in neuronal processes including cerebellar development and synaptic plasticity through Reelin signaling [38]. In humans, loss of *VLDLR* results in a non-progressive cerebellar ataxia phenotype [39] and mutations in *RELN* cause cerebellar hypoplasia [40] similar to that of the *Reeler* mouse [41]. In addition, canonical and non-canonical signaling pathways of Reelin may contribute to its neuroprotective function, for example in the regulation of tau phosphorylation in the context of Alzheimer disease pathology [42,43]. Hence, regulation of VLDLR and ApoER2 by PCSK9 may contribute to Reelin signaling and increased neuroprotection. 

Given our recent results showing higher expression of *Pcsk9* in the anterior lobules of the cerebellum where degeneration of the Purkinje neurons begins in NPC disease and the possible association with neuronal modulators ApoER2 and VLDLR, we hypothesized that decreased expression of *Pcsk9* would increase surface expression of ApoER2 and VLDLR and thus confer neuroprotection of the Purkinje neurons in the cerebellum. To test this, we characterized the *Npc1^-/-^* mouse phenotype in the presence and absence of the *Pcsk9* gene.

## 2. Results

### 2.1. Knockout of Pcsk9 Did not Significantly Alter Weight Progression in Affected Npc1^-/-^ Mice 

Mouse weights were recorded on a weekly basis from 4-11 weeks of age. Control male (*n* = 13) and female mice (*n* = 15) gained weight as expected for control male and female mice, respectively (Figure 1A,B). No obvious differences were seen between the *Pcsk9^+/+^* and *Pcsk9^+/-^* mice and these data were combined as *Pcsk9^ctrl^*. Single knockout *Npc1^-/-^*/*Pcsk9^ctrl^* male (*n* = 10) and female (*n* = 14) mice had an initial normal, but slightly reduced, weight-gain followed by weight loss starting at about 6-7 weeks of age as expected for the *Npc1^-/-^* phenotype [44,45]. In the double knockout *Npc1^-/-^*/*Pcsk9^-/-^* mice, both male (*n* = 6) and female (*n* = 7) mice were similar to the *Npc1^-/-^*/*Pcsk9^ctrl^* mice, showing a similar weight loss pattern consistent with the phenotype of *Npc1^-/-^* mutant mice alone (Figure 1A,B).

### 2.2. NPC1 Neurological Disease Progression Was Not Affected by Disruption of Pcsk9 

In order to evaluate the potential roles of *Pcsk9* in *Npc1^-/-^* mice, control, single mutant *Npc1^-/-^*/*Pcsk9^ctrl^* and double mutant *Npc1^-/-^*/*Pcsk9^-/-^* mice were assessed for disease progression on a weekly basis from 4-11 weeks of age using a composite disease severity scale [46]. No obvious differences were seen between the *Pcsk9^+/+^* and *Pcsk9^+/-^* mice and the data were combined as *Pcsk9^ctrl^*. The composite disease severity score did not show a significant difference between single mutant *Npc1^-/-^*/*Pcsk9^ctrl^* mice and double mutant *Npc1^-/-^*/*Pcsk9^-/-^* mice (Figure 1C,D). The lack of significant difference between single and double mutants was observed in both male and female cohorts (Figure 1C,D). Control mice (*Npc1^ctrl^*/*Pcsk9^-/-^*) exhibited only baseline scores that remained constant throughout the observation timeframe.

Assessment of mouse activity and motor coordination was performed in an AccuScan Activity box. Horizontal activity, walking speed and vertical activity were recorded. Rearing ability, as quantified by number of vertical movements, was used as an indication of motor coordination to complement the ledge test of the composite disease severity score. *Npc1^-/-^* mice have a well-characterized motor impairment which did not appear to be significantly altered in the absence of *Pcsk9* (Figure 2). Both *Npc1^-/-^*/*Pcsk9^ctrl^* and *Npc1^-/-^*/*Pcsk9^-/-^* cohorts had similar lower average number of vertical movements than control mice at week 9, consistent with the disease severity. This trend was consistent in both male and female mice. The other measurements of movement (horizontal activity and speed) were likewise equally affected in both *Npc1^-/-^*/*Pcsk9^ctrl^* and *Npc1^-/-^*/*Pcsk9^-/-^* groups.

### 2.3. Lifespan of Npc1^-/-^/Pcsk9^-/-^ Mice is not Significantly Altered

Gene deletion of *Pcsk9* did not appear to alter the lifespan significantly in male or female mice. The median age of death for male *Npc1^-/-^*/*Pcsk9^ctrl^* (*n* = 10) and *Npc1^-/-^*/*Pcsk9^-/-^* (*n* = 4) mice was 70.5 (range 61–80) and 75 (range 66-80) days, respectively, and was not statistically different (p = 0.209, Log-rank (Mantle–Cox) test) (Figure 3A). For female *Npc1^-/-^*/*Pcsk9^ctrl^* (*n* = 13) and *Npc1^-/-^*/*Pcsk9^-/-^* (*n* = 9) mice, the median age of death was 77 (range 63–87) and 76 (range 63–84) days, respectively (p = 0.642, Log-rank (Mantle–Cox) test) (Figure 3B). Control mice (*Npc1^ctrl^*/*Pcsk9^-/-^*) were unaffected.

### 2.4. Cerebellar Analysis of Purkinje Neurons by Immunohistochemical Staining

To exclude the possibility that increased cholesterol uptake in the liver, as expected in *Pcsk9^-/-^* mice [32], could lead to a more severe liver disease in *Npc1^-/-^*, that would mask a possible neurological benefit, we evaluated Purkinje neuron survival and liver disease in *Npc1^-/-^*/*Pcsk9^-/-^* and *Npc1^-/-^*/*Pcsk9^ctrl^* mice. Immunostaining of calbindin, as a specific marker for Purkinje neurons, in cerebellar sections from 7-week old *Npc1^+/+^*/*Pcsk9^-/-^* mice showed an intact distribution of Purkinje neurons throughout the lobules of the cerebellum (Figure 4A). This pattern is consistent with a healthy cerebellar phenotype with respect to Purkinje neurons [16] and demonstrates that lack of *Pcsk9* did not appear to affect these neurons in *Npc1^+/+^* mice although subtle changes may have occurred that were not observed. In contrast, staining from 7-week old *Npc1^-/-^*/*Pcsk9^-/-^* mice showed a significant reduction of Purkinje neurons in the anterior lobules of the cerebellum (Figure 4B), consistent with the expected Purkinje neuron loss at this age during disease progression of NPC [16]. As a control, similar staining of a 7-week old *Npc1^-/-^*/*Pcsk9^+/+^* mouse was evaluated and showed that Purkinje neuron loss appeared similar in *Npc1^-/-^* mice irrespective of the *Pcsk9* genotype (Figure 4C).

### 2.5. Serum Liver Enzymes and Cholesterol Analyses

Given the active role of PCSK9 in regulating serum cholesterol through LDLR degradation in the liver, we sought to assess liver function of 7-8-week-old *Pcsk9^-/-^* mice in the context of *Npc1*^-/-^ on the Balbc/C57BL/6 mixed background used here. We measured the activity of three liver enzymes, alkaline phosphatase (AP), alanine aminotransferase (ALT) and aspartate aminotransferase (AST), in the serum and showed significantly elevated levels in *Npc1* mutant mice (*Npc1^-/-^*/*Pcsk9^-/-^* (*n* = 9) or *Npc1^-/-^*/*Pcsk9^+/+^*(*n* = 6)), compared to the control groups (*Npc1^+/+^*/*Pcsk9^+/+^* (*n* = 4) or *Npc1^+/+^/Pcsk9^-/-^* (*n* = 8)), which were within the normal range (Figure 5). The elevated values in *Npc1^-/-^* mice are consistent with values published previously for *Npc1^-/-^* mice on a Balbc background [45]. Levels of ALT were reduced in the *Npc1^-/-^*/*Pcsk9^-/-^* mice compared to *Npc1^-/-^*/*Pcsk9^+/+^* (p = 0.031) (Figure 5B), whereas AP and AST were not significantly different (Figure 5A,C) suggesting that the diseased liver function in *Npc1^-/-^*/*Pcsk9^-/-^* mice was equivalent to or even less than the *Npc1^-/-^*/*Pcsk9^+/+^* mice at this age. Levels of AP, ALT and AST in serum from *Npc1^+/+^/Pcsk9^+/+^* mice were not significantly different from *Npc1^+/+^/Pcsk9^-/-^* mice suggesting a normal liver function phenotype in the absence of PCSK9. Total serum cholesterol in control mice (*Npc1^+/+^*/*Pcsk9^+/+^*) was 84.9 ± 12.3 mg/dL, *n* = 4. As expected for *Pcsk9^-/-^* mice [32], *Npc1^+/+^*/*Pcsk9^-/-^* mice had reduced serum cholesterol levels (68.7 ± 8.4 mg/dL, *n* = 8). In the absence of NPC1 (*Npc1^-/-^*/*Pcsk9^+/+^*), cholesterol levels were higher than control mice (142.8 ± 5.7 mg/dL, *n* = 6) which was reduced (104.9 ± 10.7 mg/dL, *n* = 9) in the *Npc1^-/-^*/*Pcsk9^-/-^* mice, demonstrating the cholesterol lowering effect of the absence of PCSK9 even in NPC1.

## 3. Discussion

The role of PCSK9 in serum cholesterol regulation has been well established. Indeed, since its discovery in 2003, our current knowledge represents the complement of comprehensive basic and translational science, leading ultimately to an effective treatment of hypercholesterolemia [30]. However, until more comprehensive studies on PCSK9 are undertaken to explore its role in the central nervous system (CNS), treatment of PCSK9-regulating drugs may affect the CNS in unseen and unwanted ways. 

The exact role of PCSK9 in the CNS remains unknown. However, due to its well characterized cell biological mechanism in regulating the levels of the LDL receptor in the periphery, it is expected to function similarly in the brain. Studies have shown it can bind and regulate the degradation of the VLDL and ApoE2 receptors in a similar manner to that of the LDL receptor. In addition, since mature neurons receive their cholesterol from astrocytes through VLDLR and ApoER2, it is predicted that PCSK9 could contribute to brain cholesterol metabolism by regulating these receptors. In the adult mouse brain, it has been reported that PCSK9 is not involved in LDLR degradation in the cortex and hippocampus [36]. However, *Pcsk9* is increased in the dentate gyrus after transient cerebral ischemia and in a mouse model of hyperlipidemia where neuronal apoptosis occurs in the hippocampus [47,48]. In both cases, reduced LDLR levels and increased neuronal apoptosis in these tissues appear to correlate with the increase in *Pcsk9*. This suggests that in the context of disease or injury, levels of PCSK9 may regulate neuronal apoptotic signaling pathways. Based on these observations we therefore sought to test whether PCSK9 plays a role in NPC1, a disease of cholesterol metabolism. 

Our studies however have shown that PCSK9 does not appear to play a role in NPC1 pathology even though we found expression differences in the lobules susceptible and resistant to Purkinje neuron death [16]. In all parameters measured (disease severity score, weight, lifespan, selected motor functions and Purkinje neuron staining and serum liver enzymes), we did not discern an obvious difference in NPC1 disease progression in mice with or without PCSK9. Perhaps the levels of expression of PCSK9 are too low in the adult cerebellum to affect receptor levels especially in the Purkinje neurons layer, or perhaps specificity and sensitivity of binding to VLDLR or ApoER2 are different in vivo in the context of a cholesterol trafficking defect.

While further studies could be undertaken to characterize the molecular effects in the brain, we have decided that in the context of NPC1 disease outcome, this would not be informative. We present this work here to allow colleagues in the field to assess this aspect of NPC1 and conclude as we do that PCSK9 does not play a significant role in NPC1 disease.

## 4. Materials and Methods

### 4.1. Animal Maintenance

All animal work was reviewed and approved by the *Eunice Kennedy Shriver* National Institute of Child Health and Human Development (NICHD) Animal Care and Use Committee under protocol #18-002 (28 May 2018). Heterozygous *Npc1^+/-^* mice (BALB/cNctr-Npc1m1N/J) and B6;129S6-*Pcsk9^tm1Jdh/J^* (*Pcsk9^-/-^)* were purchased from The Jackson Laboratory (Bar Harbor, ME, USA). *Npc1^+/-^* mice were crossed with *Pcsk9^-/-^* mice to obtain double heterozygous mutants (*Npc1^+/-^/Pcsk9^+/-^*). These mice were then inter-crossed to generate double mutant mice (*Npc1^-/-^/Pcsk9^-/-^*) and the corresponding control mice. Pups were weaned 3 weeks post-birth and subsequently had free access to food and water. Genotypes were confirmed by PCR using DNA extracted from ear clips and the genotyping protocol reported by The Jackson Laboratory (protocol ID=27858). A humane end of life timepoint was determined in conjunction with the animal facility veterinarian, by assessing severity of motor impairment, at which point the mice were euthanized and tissues analyzed as necessary.

### 4.2. Disease Severity Scoring

After weaning, mice were assessed weekly for weight and disease severity using a standardized scoring system modified for NPC1 [46]. The disease severity score consisted of 5 components: Grooming, gait, kyphosis, hindlimb clasp, and ledge test, with each observation scored from 0-3; 0 being the least severe and 3 indicating a severely affected phenotype. Upon entry to the procedure room, mice were allowed to acclimate to the new environment for 15 minutes. For the assessment, a mouse was randomly selected and weighed. The mouse was then placed into a new cage and assessed for disease phenotype. A composite disease severity score was calculated by adding the scores for each assessment. Evaluations were performed by two observers and mouse identification was revealed post-assessment. For the ledge test, mice were placed on the ledge of the cage and assessed for their ability to walk around the ledge with coordinated use of hindlimbs, proper balance, and ability to dismount effortlessly. 

### 4.3. AccuScan Activity Assessment

One of the measures of motor coordination is the ability of the mice to rear up on their hind legs. We therefore measured the rearing activity of the mice in a 400 mm X 400 mm AccuScan Activity box (AccuScan Instruments Inc., Columbus, OH, USA), weekly from 5-10 weeks of age. This box contains 16 infra-red laser beams that cross the area of the box at 1-inch intervals and 1 inch above the floor to measure horizontal activity. An additional set of 16 lasers are active at 3.5 inches above the floor to measure vertical activity. Mice were initially brought to the behavioral room and allowed to acclimate for 30 minutes. In the room, ambient light and sound were constant and the researcher remained silent during data acquisition. Individual mice were placed in the chambers and activity was measured automatically by beam break counts collected by computer. Horizontal and vertical counts were recorded for three contiguous 5-minute periods. Total movements were calculated and reported for the combined second and third 5-minute periods (10 minutes).

### 4.4. Immunohistochemical Analysis of Cerebellar Tissue from Mutant Mice

Mice were euthanized at 7 weeks of age and transcardially perfused with room temperature phosphate buffered saline (PBS). The whole brain was dissected immediately and fixed in 4% paraformaldehyde (PFA) in PBS, pH 7.4, 4 °C, for 24 h followed by cryoprotection in 30% sucrose until sectioning. Brains were cryostat-sectioned parasagitally (20 μm) and sections were collected in PBS containing 0.25% Triton X-100. The floating sections were blocked for 30 min in 10% normal goat serum and then incubated overnight at 4 °C with rabbit anti-Calbindin (1:400; Cell Signaling, Danvers, MA, USA) in 0.25% Triton X-100/PBS. After washing, the sections were incubated with Alexa Fluor 594 goat anti-rabbit IgG, (1:1000, Thermo Fisher Scientific, Waltham, MA, USA). Nuclei were stained with Hoechst 33342, Trihydrochloride, Trihydrate I (1:1000, Invitrogen, Carlsbad, CA, USA). Images were taken on a Zeiss Axio Observer Z1 microscope (Zeiss, Germany).

### 4.5. Serum Liver Enzymes and Cholesterol Analyses

To assess liver function, mice were euthanized by CO_2_ and blood obtained by cardiac puncture. The samples were allowed to coagulate at room temperature for 45 min and then centrifuged for 10 min at 2000 rpm, 4 °C. The serum was saved at –20 °C until sent for analysis. Sera from control and affected mice (7–8 weeks of age) were sent to the Department of Laboratory Medicine, Clinical Center, NIH for hepatic and lipid panel screens. The liver enzymes assayed were alkaline phosphatase (AP), alanine transaminase (ALT), and aspartate transaminase (AST) using the following Roche Diagnostics reagent kits; AP #03333752 190; ALT #20764957 322; AST #20764949 322 and the lipids analyzed were total cholesterol using reagent kit #03039773 190. All assays were analyzed by Roche/Hitachi cobas c 311/501 analyzers (Roche Diagnostics GmbH, Mannheim, Germany). 

### 4.6. Statistical Analysis

All statistical analyses were performed using the GraphPad PRISM software, version 8.3.0 (San Diego, CA, USA, www.graphpad.com). Data were analyzed by one-way ANOVA followed by Tukey’s multiple comparisons test. The log rank Mantle–Cox test for significance was used for the Kaplan–Meyer survival curve. Statistical significance was set to *p* < 0.05.

## Figures and Tables

**Figure 1 ijms-21-02430-f001:**
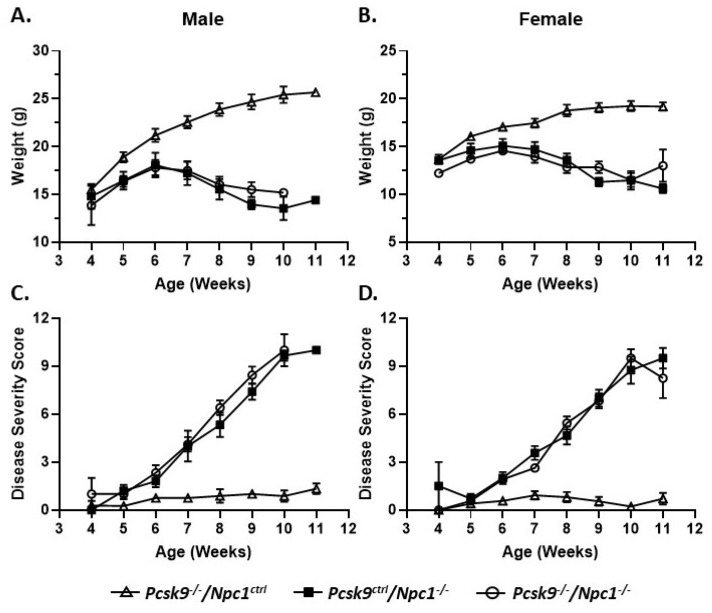
Weight and disease progression in proprotein convertase subtilisin/kexin type 9 deficient mice (*Pcsk9^-/-^*) in the context of NPC1 disease. Male (**A**) and female (**B**) mice were weighed weekly. *Pcsk9^-/-^/Npc1^ctrl^* mice had a normal weight gain profile. *Npc1* mutant mice begin to lose weight at ~6 weeks of age and continued thereafter. There was no significant difference observed between *Pcsk9^ctrl^/Npc1^-/-^* and *Pcsk9^-/-^/Npc1^-/-^* mice. Male (**C**) and female (**D**) mice were assessed on a weekly basis for disease severity: grooming, gait, extent of kyphosis, hindlimb clasp and ability to walk along the ledge of the cage. As expected, *Pcsk9^-/-^/Npc1^ctrl^* mice displayed negligible scores throughout the course of the assessments. The *Npc1* mutant mice exhibited an increase in disease severity starting at ~6 weeks of age and continued thereafter. No significant difference was observed between *Pcsk9^ctrl^/Npc1^-/-^* and *Pcsk9^-/-^/Npc1^-/-^* groups.

**Figure 2 ijms-21-02430-f002:**
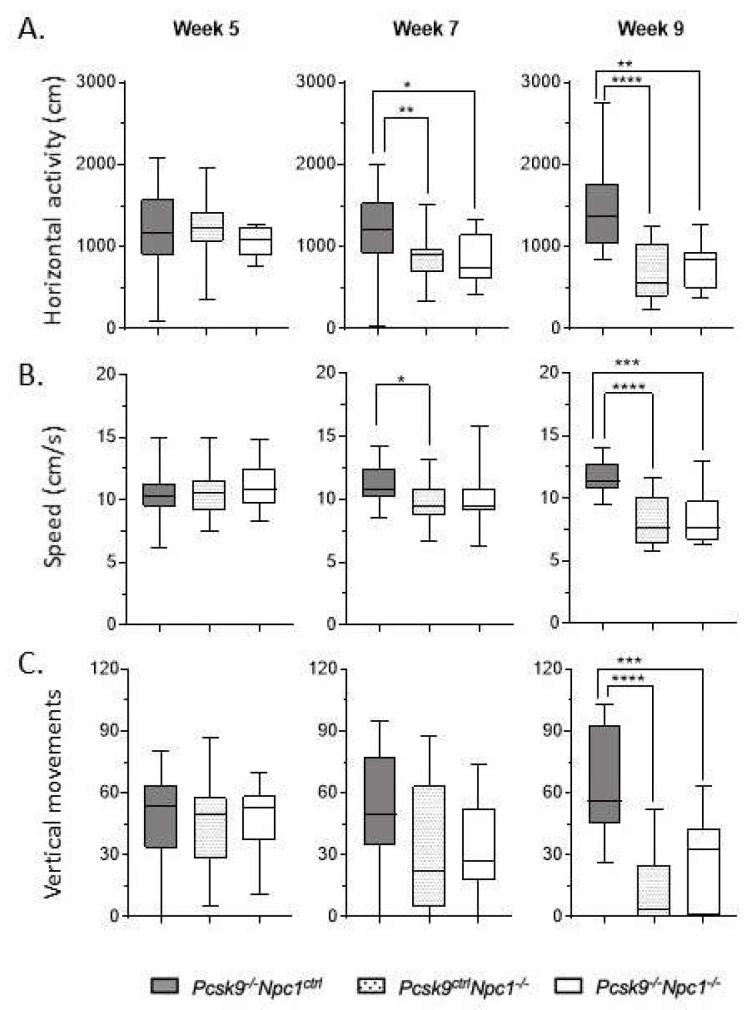
Motor coordination in *Pcsk9^-/-^* mice in the context of NPC1 disease. Using the AccuScan activity box, the horizontal activity (**A**), walking speed (**B**), and vertical activity (**C**) of the mice were recorded on a weekly basis. Results from 5, 7 and 9 weeks are plotted here. As expected, *Pcsk9^-/-^/Npc1^ctrl^* showed consistent measurements of all activity across all timepoints. In contrast, *Pcsk9^ctrl^/Npc1^-/-^* and *Pcsk9^-/-^/Npc1^-/-^* groups displayed decreased horizontal activity and walking speed starting at week 7, which was worse and more significant at 9 weeks. The average number of vertical movements of both *Pcsk9^ctrl^/Npc1^-/-^* and *Pcsk9^-/-^/Npc1^-/-^* groups was significantly decreased by week 9. Asterix indicate significance levels, * *p* < 0.05; ** *p* < 0.01; *** *p* < 0.001; **** *p* < 0.0001.

**Figure 3 ijms-21-02430-f003:**
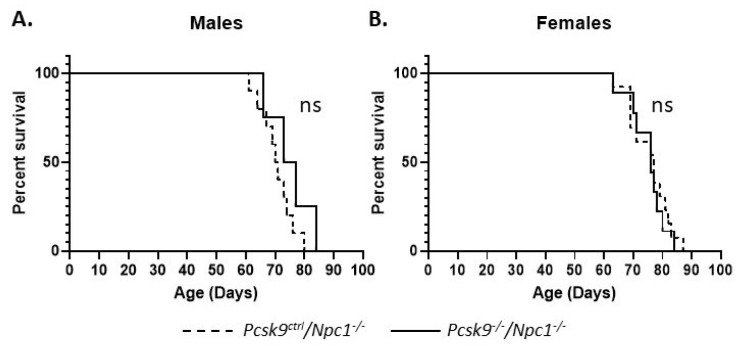
Mouse survival rates in *Pcsk9^-/-^* mice in the context of NPC1 disease. Male and female mutant mice were assessed for survival. The end of life timepoint was assessed based on severity of motor impairment. No significant difference in survival was observed between male (**A**) or female (**B**) *Pcsk9^ctrl^/Npc1^-/-^* and *Pcsk9^-/-^/Npc1^-/-^* mice.

**Figure 4 ijms-21-02430-f004:**
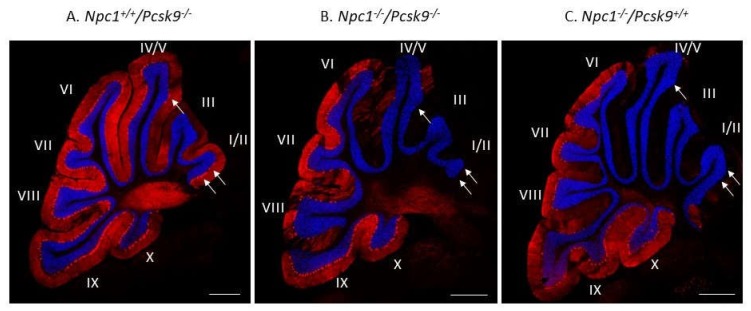
Purkinje cell loss in *Pcsk9^-/-^* mice in the context of NPC1 disease. Cerebellar sections from 7-week old *Pcsk9^-/-^/Npc1^+/+^* (**A**), *Pcsk9^-/-^/Npc1^-/-^* (**B**) and *Pcsk9^+/+^/Npc1^-/-^* (**C**) mice were analyzed by indirect immunohistochemical staining. Purkinje neurons were stained for calbindin (red) and counterstained with DAPI (blue) to distinguish the granule cell layer from the molecular layer. In *Pcsk9^-/-^/Npc1^+/+^* sections (**A**), Purkinje neurons were found generally in all lobules and especially in lobule I/II. In *Pcsk9^-/-^/Npc1^-/-^* (**B**) and *Pcsk9^+/+^/Npc1^-/-^* (**C**) sections, a marked decrease of calbindin staining in the anterior lobules (I-IV/V) indicated Purkinje neuron death consistent with the level of NPC1 disease at this age. No obvious difference was observed in the calbindin staining pattern between *Pcsk9^-/-^/Npc1^-/-^* and *Pcsk9^+/+^/Npc1^-/-^* mice, suggesting that deletion of *Pcsk9* does not appear to alter Purkinje cell death progression in the context of NPC1 disease. Bar = 200 μm. Arrows point to Purkinje neuron cell bodies in panel A, and their absence in panels B, C.

**Figure 5 ijms-21-02430-f005:**
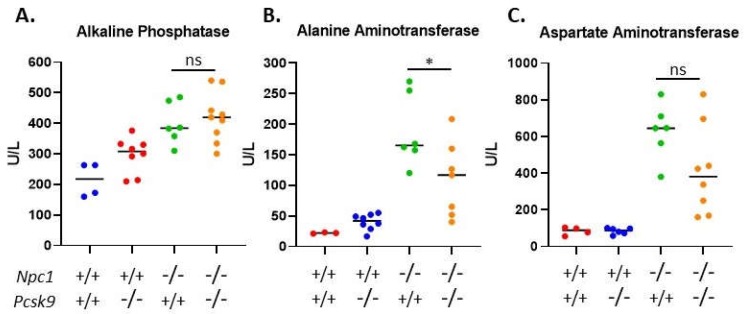
Serum liver enzyme levels in *Pcsk9^-/-^* mice in the context of Niemann–Pick disease, type C1 (NPC1) disease. In order to assess liver function, serum was collected from control and mutant mice at 7–8 weeks of age. All three enzymes, (**A**) alkaline phosphatase (AP), (**B**) alanine aminotransferase (ALT) and (**C**) aspartate aminotransferase (AST), were elevated in the *Npc1^-/-^* mice compared to control mice. The levels of AP and AST between *Pcsk9^-/-^/Npc1^-/-^* and *Pcsk9^+/+^/Npc1^-/-^* were not statistically different, whereas the ALT was slightly reduced in the *Pcsk9^-/-^/Npc1^-/-^* mice. These results demonstrate that liver function was affected similarly in *Npc1^-/-^* mice in the presence or absence of PCSK9.

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
