# Peer review of "Evaluation of the Potential Role of Proprotein Convertase Subtilisin/Kexin Type 9 (PCSK9) in Niemann–Pick Disease, Type C1"

_ijms, 2020, doi:10.3390/ijms21072430_

Round 1

Reviewer 1 Report

The present work addresses a very important question: what are modifiers of Niemann-Pick disease that decide upon life or death of cells following NPC1 dysfunction. They may ultimately serve as much wanted therapeutic targets. Based on previous work, the authors tested a prime candidate, PCSK9, which has become a hot target for cholesterol-related diseases. To this end, the authors compared the outcome of NPC1-deficiency in the presence or absence of PCSK9, and found no differences in several key parameters (body weight, life-span, density of cerebellar neurons). These obviously negative results are very important for the field, as they will stimulate further studies into the regulation of lipid metabolism in neurons and their reaction to dysfunction of key components such as NPC1. The ms is very complete in terms of experiments, the results have been analysed and presented in an exemplary manner. However, the authors should consider the following points for a minor revision:

- Lines 80-95: Here, the Introduction could be shortened a bit. The relevance of reelin signaling in the context of the authors' study is not really clear.

- Lines 169-170 The sentence "lack of Pcsk9 did not appear to affect these neurons in Npc1+/+ mice" should be replaced by a more cautious statement. Evidently, the authors cannot exclude albeit subtle changes in the functional properties of the cells. The authors should make sure to avoid such statements throughout the ms.

- Lines 187-204: This paragraph merits a separate headline, it does not fit to "2.4 Cerebellar analysis..."

- Lines 214-216: The sentence should be replaced, it is not really clear what the authors intend to convey.

- Lines 219-241: The discussion could be shortened. This paragraph for example is not really related to the authors' work (see Introduction). Instead, the authors should discuss why PCSK9 does not modify the outcome of NPC1 deficiency. Here, the following points can be considered: Could this be due to its reliance on lysosomal function, i.e. can PCSK9 cleave lipoprotein receptors at all in the absence of NPC1, which by itself causes severe lysosomal dysfunction? If not, it's absence from mice would not make a difference indeed. Does the genetic background affect the phenotype of PCSK9 deficiency in mice?

- Line 277: Correct the expression "is the ability for the mice".

- Lines 307-310: The authors used statistical tests other than ANOVA, this should be mentioned.

- Throughout the ms, the terms immunofluorescence and immunostaining etc. should be replaced by the correct term "immunohistochemical staining"

- Fig. legends: The term "in the context of NPC1" is unclear. Does this mean NPC1 disease? If so, this should be corrected, and the abbreviation should be defined.

Author Response

The present work addresses a very important question: what are modifiers of Niemann-Pick disease that decide upon life or death of cells following NPC1 dysfunction. They may ultimately serve as much wanted therapeutic targets. Based on previous work, the authors tested a prime candidate, PCSK9, which has become a hot target for cholesterol-related diseases. To this end, the authors compared the outcome of NPC1-deficiency in the presence or absence of PCSK9, and found no differences in several key parameters (body weight, life-span, density of cerebellar neurons). These obviously negative results are very important for the field, as they will stimulate further studies into the regulation of lipid metabolism in neurons and their reaction to dysfunction of key components such as NPC1. The ms is very complete in terms of experiments, the results have been analysed and presented in an exemplary manner. However, the authors should consider the following points for a minor revision:

 Thank you for taking the time to review our work. Your comments and suggestions have made the manuscript clearer for the reader. We have addressed all your points carefully as outlined in our responses.

- Lines 80-95: Here, the Introduction could be shortened a bit. The relevance of reelin signaling in the context of the authors' study is not really clear.

 Our apologies for the lack of clarity in connecting the relevance of reelin signaling to our study. If PCSK9 can regulate the reelin receptors in the brain, we considered that introducing what reelin does and describing defects in reelin signaling that give rise to cerebellar ataxia, might help the reader draw on the importance and add rationale for why we did the study.

As requested, we have shortened this paragraph a bit. We deleted the sentence about the Dab1 and Src family kinases (deleted at new line 95, Revision 1). However, we believe it is still important to introduce reelin signaling in connection to PCSK9.

- Lines 169-170 The sentence "lack of Pcsk9 did not appear to affect these neurons in Npc1+/+ mice" should be replaced by a more cautious statement. Evidently, the authors cannot exclude albeit subtle changes in the functional properties of the cells. The authors should make sure to avoid such statements throughout the ms.

We agree with your comment here. We had considered our statement to be sufficiently cautious so as not to over interpret our results. However, we have amended the statement to include that subtle changes may have occurred that were not measured or observed in our experiments (line 176, Revision 1). We have also made similar edits throughout the manuscript (line 180, 191, Revision 1).

- Lines 187-204: This paragraph merits a separate headline, it does not fit to "2.4 Cerebellar analysis..."

Thank you, we have now given this paragraph/section a separate heading. “2.5 Serum liver enzymes and cholesterol analyses”. (line 194, Revision 1)

- Lines 214-216: The sentence should be replaced, it is not really clear what the authors intend to convey.

This sentence was simply meant as an opening statement of the Discussion to emphasize that the study of PCSK9 has been well documented and was meant also to pay tribute to a field that progressed from initial observation to functional treatment. It is an impressive body of work.

We have edited this opening statement to better convey that further study of PCSK9 is required to fully understand its possible roles in the CNS (line 225, Revision 1)

- Lines 219-241: The discussion could be shortened. This paragraph for example is not really related to the authors' work (see Introduction). Instead, the authors should discuss why PCSK9 does not modify the outcome of NPC1 deficiency. Here, the following points can be considered: Could this be due to its reliance on lysosomal function, i.e. can PCSK9 cleave lipoprotein receptors at all in the absence of NPC1, which by itself causes severe lysosomal dysfunction? If not, it's absence from mice would not make a difference indeed. Does the genetic background affect the phenotype of PCSK9 deficiency in mice?

You are correct, this section in the Discussion is not directly related to the content of the manuscript. We were attempting to balance the Introduction section and the Discussion section to emphasize the importance of PCSK9 and why we did the study in the first place. We have now transferred a paragraph from the Discussion and edited it into the Introduction (line 79, Revision 1). While this increases the length of the Intro, it makes the Discussion more concise. We also added new discussion points as requested (line 245, Revision 1).

Since its discovery, the question as to PCSK9s ability to act as a proteinase has always been an interesting question. To date, no evidence exists to indicate that it functions as a proteinase except in its very first intra-molecular processing if its prodomain. This domain remains very tightly associated with the catalytic domain, inhibiting it. It is certainly possible that under certain conditions in vivo the enzyme may be found to function as a proteinase, however, it would only be speculation at this point.

- Line 277: Correct the expression "is the ability for the mice".

This has now been corrected.

- Lines 307-310: The authors used statistical tests other than ANOVA, this should be mentioned.

We have now added in this section that the Mantle-Cox log rank test was used for the Kaplan-Meyer survival curve statistical analyses (line 312-3, Revision 1).

- Throughout the ms, the terms immunofluorescence and immunostaining etc. should be replaced by the correct term "immunohistochemical staining"

We have changed this throughout the manuscript to immunohistochemical staining.

- Fig. legends: The term "in the context of NPC1" is unclear. Does this mean NPC1 disease? If so, this should be corrected, and the abbreviation should be defined.

Yes, it was meant to indicate Niemann-Pick Type C1 disease. We have now corrected this in the figure legends. NPC1 is already listed in the abbreviations section.

Reviewer 2 Report

This is an interesting study showing the effects of PCSK9 in neurodegeneration induced by NPC1-deficiency. The authors investigated the effects of PCSK9/NPC1 deficiency on disease severity scoring, motor assessments, lifespan and cerebellar Purkinje cell staining in genetically-engineered mice. The authors concluded that PCSK9 does not play an apparent role in NPC1 disease progression. The study is technically sound, the main conclusions are supported by data but some issues should be addressed by the authors.

My main concern is that the authors appears not to take into account the PCSK9/NPC1-mediated potential changes in LDLR, VLDLR and apoE2 receptor and no data are shown about these relevant receptors and/or any compensatory mechanisms.

The authors indicate “To exclude the possibility that increased cholesterol uptake in the liver…” but they do not determine liver cholesterol rather hepatic function and serum cholesterol (this is not the same). This point should be addressed rather by determining liver cholesterol (or alternatively by liver lipid staining although Oil red O also would detect triglycerides).

Methods for determining hepatic parameters and serum cholesterol should be included.

Author Response

This is an interesting study showing the effects of PCSK9 in neurodegeneration induced by NPC1-deficiency. The authors investigated the effects of PCSK9/NPC1 deficiency on disease severity scoring, motor assessments, lifespan and cerebellar Purkinje cell staining in genetically-engineered mice. The authors concluded that PCSK9 does not play an apparent role in NPC1 disease progression. The study is technically sound, the main conclusions are supported by data but some issues should be addressed by the authors.

Thank you for taking the time to review our manuscript. We appreciate your comments and suggestions and understand that obtaining the results from your suggestions would make our study more complete.

We are currently not in a position to be able to address these with experiments since our lab is under lock down due to the issues surrounding COVID-19 and the safety of patient and personnel of the Clinical Center. Please see our responses below. While not ideal, we hope our responses are adequate, given the circumstances.

My main concern is that the authors appears not to take into account the PCSK9/NPC1-mediated potential changes in LDLR, VLDLR and apoE2 receptor and no data are shown about these relevant receptors and/or any compensatory mechanisms.

This is a very good point, however, during our study when we were sure of the apparent lack of a difference in disease progression, whether PCSK9 was present or not, we decided then not to pursue this, but to focus our time and resources on completing the other analyses in our study. We agree that there may be subtle changes at the molecular level, but for our purpose, such changes had no apparent effect on outcome. As it stands now, however, we currently cannot perform these experiments due to the lock-down of our laboratory.

The authors indicate “To exclude the possibility that increased cholesterol uptake in the liver…” but they do not determine liver cholesterol rather hepatic function and serum cholesterol (this is not the same). This point should be addressed rather by determining liver cholesterol (or alternatively by liver lipid staining although Oil red O also would detect triglycerides).

This is also a very good point; however, may we suggest that our justification that liver function is more important from a clinical standpoint and should suffice for us to interpret that liver disease is at least the same in the double KO mouse as the single Npc1 mutant. Cholesterol storage in the liver may or may not be different between the mutant genotypes, but in any case, we interpret that the lack of PCSK9 does not increase the liver defect in NPC1 disease. We request that our experiments can stand as is without further analysis of liver cholesterol analysis as it would not change our overall interpretation of the study.

Methods for determining hepatic parameters and serum cholesterol should be included.

The analysis of the serum was carried out by the Dept. of Laboratory Medicine, a component of the Clinical Research Center, NIH. We are awaiting a response from the Dept. with the information about the methods used. Due to the Clinical Research Center work restrictions, it may take a little longer to get a reply back with the information, but likely they will be standard and routinely used commercial kits.

Round 2

Reviewer 2 Report

I understand the difficulties to conduct some experiments in the context of COVID-19, but at least commercial kits used for liver and lipid parameters should be included.

Author Response

Thank you for your understanding. I have now added the information about the assays for serum liver enzymes and total cholesterol in the Materials and Methods section.

“The liver enzymes assayed were alkaline phosphatase (AP), alanine transaminase (ALT) and aspartate transaminase (AST) using the following Roche Diagnostics reagent kits; AP #03333752 190; ALT #20764957 322; AST #20764949 322 and the lipids analyzed were total cholesterol using reagent kit #03039773 190. All assays were analyzed by Roche/Hitachi cobas c 311/501 analyzers.”